# Influencing Activity of Bats by Dimly Lighting Wind Turbine Surfaces with Ultraviolet Light

**DOI:** 10.3390/ani12010009

**Published:** 2021-12-21

**Authors:** Paul M. Cryan, Paulo M. Gorresen, Bethany R. Straw, Syhoune (Simon) Thao, Elise DeGeorge

**Affiliations:** 1U.S. Geological Survey (USGS), Fort Collins Science Center, Fort Collins, CO 80526, USA; bstraw@usgs.gov; 2Hawaii Cooperative Studies Unit, University of Hawaii at Hilo, Hilo, HI 96720, USA; mgorresen@usgs.gov; 3USGS Pacific Island Ecosystems Science Center, Hawaii Volcanoes National Park, Hilo, HI 96718, USA; 4U.S. Department of Energy, National Renewable Energy Laboratory, National Wind Technology Center, Boulder, CO 80007, USA; syhoune.thao@nrel.gov (S.T.); Elise.DeGeorge@nrel.gov (E.D.)

**Keywords:** artificial illumination at night (ALAN), Chiroptera, behavior, deterrence, fatality, wildlife, wind energy

## Abstract

**Simple Summary:**

Bats often fly near wind turbines. The fatalities associated with this behavior continue to be an issue for wind energy development and wildlife conservation. We tested an experimental method intended to reduce bat fatalities at the wind turbines. We assumed that bats navigate over long distances at night by dim-light vision and might be dissuaded from approaching artificially lit structures. For over a year, we experimentally lit wind turbines at night with dim, flickering ultraviolet (UV) light while measuring the presence and activity of bats, birds, and insects with thermal-imaging cameras. We detected no statistical differences in the activity of the bats, insects, or birds at a test turbine when lit with UV light compared with that of unlit nights. Additional experiments to test this or other possible bat-deterrence methods may benefit from considering subtle measures of animal response that can provide useful information on the possible behavioral effects of fatality-reduction experiments.

**Abstract:**

Wind energy producers need deployable devices for wind turbines that prevent bat fatalities. Based on the speculation that bats approach turbines after visually mistaking them for trees, we tested a potential light-based deterrence method. It is likely that the affected bats see ultraviolet (UV) light at low intensities. Here, we present the results of a multi-month experiment to cast dim, flickering UV light across wind turbine surfaces at night. Our objectives were to refine and test a practical system for dimly UV-illuminating turbines while testing whether the experimental UV treatment influenced the activity of bats, birds, and insects. We mounted upward-facing UV light arrays on turbines and used thermal-imaging cameras to quantify the presence and activity of night-flying animals. The results demonstrated that the turbines can be lit to the highest reaches of the blades with “invisible” UV light, and the animal responses to such experimental treatment can be concurrently monitored. The UV treatment did not significantly change nighttime bat, insect, or bird activity at the wind turbine. Our findings show how observing flying animals with thermal cameras at night can help test emerging technologies intended to variably affect their behaviors around wind turbines.

## 1. Introduction

Prior to the late 1990s, bat fatalities resulting from collisions with human-made structures were uncommon [1]. Unprecedented numbers of bat fatalities have followed the deployment of tall, monopole wind turbines in recent decades [2,3,4]. Reports of bat collision fatalities continue to accumulate from wind energy facilities all over the world and contribute to wind turbines now ranking highest among the currently known sources of mass mortality in bats [5,6,7,8,9,10]. This bat/turbine problem disproportionately affects certain species, potentially threatening their populations at scales ranging from entire continents to remote islands (e.g., the North American hoary bat [*Lasiurus cinereus*] and the Hawaiian hoary bat [*L. semotus*] [8,11]). Predicting how the deaths of individual bats at wind turbines relate to status and trends in their constituent populations remains a major challenge [12]; yet, current evidence suggests that the populations of bats most affected by turbines could become extinct [8,13]. Wind-energy mortality can disrupt migratory connectivity and further impact species adversely affected by other natural stressors, such as disease [14]. Lack of understanding about how to predict the bat fatality risk at wind turbines based on surrounding landscape features or the bat activity levels at sites before the wind turbines were built exacerbates conservation and facility-siting challenges [9,15,16,17,18,19]. Widespread bat fatalities impede wind energy operations, and we lack proven methods for preventing the fatalities that are free of other environmental, economic, or energy availability effects.

Two strategies have emerged over recent decades for trying to remedy the bat/turbine problem: curtailment and deterrence. The first involves altering the operation of wind turbines to reduce the chances of bats colliding with the moving blades. Curtailing, or feathering the turbines blades to slow the rotation of the rotor under certain conditions, became the most adopted operational change for reducing fatalities after studies demonstrated its efficacy under low-wind conditions in late summer and autumn [20,21]. However, curtailment can reduce energy production, limiting the current and future installed capacity for wind energy and has not fully prevented incidental bat fatalities. Over time, scientists have explored a spectrum of curtailment strategies that utilize a wide-ranging set of environmental variables, sometimes including sensor systems for detecting bat presence, to trigger curtailment [22,23,24]. The second strategy for addressing the bat/turbine problem involves using turbine-mounted devices to broadcast aversive stimuli, such as ultrasonic noise that deters the bats from approaching the airspace around the turbines [25,26,27]. The limitations of this method include the extent of the space around the turbines that can be effectively treated and the differential responses among species of bats. Another device-based method for deterring animals from colliding with human-made structures involves enhancing or aversively altering a structure’s visible appearance to approaching individuals. Researchers developed this method for birds and other types of structures, such as building windows and power lines, but to our knowledge have not tested it on bats and wind turbines [28,29,30]. One study altered the surface texture of a wind turbine to test whether echolocating bats approached it less, but that effort aimed to change the way echolocating bats perceive turbines through sound [31]. Changing the way bats visually perceive wind turbines from a distance might help keep them at safe distances.

Operators of wind energy facilities could benefit from additional device-based methods that further reduce bat fatalities, while working at further distances, greater speeds, and on more species than the existing systems. Light may be a preferable physical medium with which to signal or dissuade bats as they approach wind turbines because bats can perceive light at greater distances and speeds than is physically possible with sound [32,33,34,35,36]. Vision seems to be the primary means by which bats navigate and orient across longer distances [37,38]. Bats are probably able to see very tall or large structures at night in ambient dim light, and certain tree-roosting bats may visually detect trees, wind turbines, and illuminated tall structures at night from very far away [38,39,40]. Species of bats found dead under turbines most often tend to be those that live in trees most or all of the year and travel the furthest among seasons. For example, 4 of the 45 species of bats that occur in the United States and Canada comprise the vast majority (>95%) of documented fatalities at turbines there [1,6]. Unique aspects of the distribution, physiology, and behaviors of these turbine-susceptible species as they relate to fatalities at wind turbines led to extensive speculation about the causal links between their susceptibility and the structural similarities of wind turbines and trees [3,39,41,42,43,44,45].

In addition to generally finding their way around landscapes at night, bats use vision and ambient light to locate roosts in trees and other structures [46]. Bats have lower visual acuity (sharpness of vision) than humans and birds but can see well in very dim light that appears dark to humans—bat vision generally remains functional at illumination intensities below those where humans and most birds can see [47,48,49,50,51]. Regardless of probably being able to see wind turbines at night, bats may not be able to discriminate the silhouettes of large trees from the cylindrical towers and branch-like blades of wind turbines under such conditions [3,39,40,41,43,44]. Prior to the deployment of large wind turbines on the landscape, few other tall structures so closely resembled large trees—the preponderance of bat species that live in and around trees among the fatalities at wind turbines does not appear coincidental. Certain types of bats might visually mistake turbines for tall, prominent trees around which they are likely to have evolved behaviors important to individual survival and species persistence [3,39,40,41,43,44].

We speculate that we might prevent bats from approaching wind turbines by subtly lighting these dynamic structures in ways that visually reduce the likelihood that approaching bats will perceive them as trees in silhouette. However, illuminating turbine surfaces with light visible to human observers is likely to be unacceptable due to existing concerns about the visual obtrusiveness of turbines at night, as well as possibly attracting and/or confusing nocturnally migrating birds and insects. We therefore decided to experiment with illuminating turbine surfaces in a spectral range of light that bats can see but humans cannot and at lower intensities than can probably be perceived by birds [51]. In general, the vision of bats spans wavelengths from about 360 to 680 nm [52,53]. Illumination in the near-ultraviolet (UV) spectrum of light (approximately 300 to 400 nm) particularly interested us as a method for invisibly signaling bats near wind turbines without humans, birds, and other animals perceiving the cue. The evidence gathered over the past decade strongly indicates that many bats, including most species known to be affected by wind turbines in North America and Europe, have the visual anatomy to perceive UV light [53,54,55,56,57,58,59]. These genetic studies of multiple species, and a more limited number of laboratory and field experiments with captive or free-ranging bats, demonstrate that many species of echolocating bats might indeed be able to see UV light (e.g., [60,61,62]). A field experiment to reduce Hawaiian hoary bat activity by flickering dim UV light on trees in a natural foraging environment demonstrated a decrease in bat vocalization and proximity to the illuminated area concomitant with a slight increase in insect activity [63].

In summary, we developed and tested a potential light-based method of reducing bat fatalities at wind turbines that stems from speculation that bats approach wind turbines that they visually mistake for trees. Our objectives were to design and test a practical system for dimly UV-illuminating turbine surfaces while determining whether the experimental UV treatment influenced the activity of bats, birds, and insects. Here, we present the results of a multi-month experiment to cast dim, flickering UV light across the surfaces of wind turbines at night.

## 2. Materials and Methods

### 2.1. Study Site

We conducted this study from 15 August 2018 to 31 October 2019 on two wind turbines at the National Wind Technology Center at the National Renewable Energy Laboratory (NREL) in Boulder, Colorado (Figure 1). At least 5 species of 4 bat genera are known to occur at the site, including the big brown bat (*Eptesicus fuscus*), the fringed myotis (*Myotis thysanodes*), the silver-haired bat (*Lasionycteris noctivagans*), the hoary bat (*Lasiurus cinereus*), and the eastern red bat (*Lasiurus borealis*), as well as other species of *Myotis* that could not be reliably classified by their echolocation calls [64]. We chose these turbines for the study because they were conveniently located and accessible for this research, which required year-round observation, technical assistance from facility staff, and maintenance access. The first wind turbine, hereafter the “North Turbine,” was the 1.5-MW unit (39.9121° N, 105.2200° W; Model GE 1.5sle, General Electric Renewable Energy, Schenectady, NY, USA) previously studied by Goldenberg et al. [65]. The second turbine, hereafter the “South Turbine”, was a different make and model (Model SWT-2.3-108, Siemens, Munich, Germany) maintained and operated by an energy company on-site under a cooperative research and development agreement with the NREL. The North and South Turbines had tapered monopoles that were 80 m tall, had different-shaped nacelles housing their generators at the top of the monopoles, and had 77 m and 108 m rotor diameters, respectively. The South Turbine was approximately 700 m south-southeast of the North Turbine. Three additional wind turbines of different makes, models, and sizes also operated within several hundred meters between or to the west of the studied turbines. Urbanized and arid rangeland, transected by several drainages to the north, east, and south, surrounded the turbines and transitioned into the foothills of the Rocky Mountains approximately 5 km to the east (Figure 1).

### 2.2. Ultraviolet Lighting System

To try and elicit behavioral responses by bats to dimly illuminated turbine surfaces, we equipped each of the two wind turbines with an experimental lighting system (Figure 2), consisting of 12 illuminator units powered and controlled through aircraft-grade electrical cables with 13-V, 20-amp DC power supplies (Model W13NT2000U, Acopian Technical Company, Easton, PA, USA). A programable 365 day digital timer controlled each lighting system (Model ET90000, Intermatic Incorporated, Spring Grove, IL, USA). We housed each power supply and timer within a safety enclosure inside the tower at the base of each turbine and only accessed them at approximately bi-monthly intervals for maintenance. Our illuminator was custom designed and fabricated for the study to cast UV light with a peak wavelength of 365 nm and a power density of approximately 1 µW/cm^2^ over a circular region of a 20 m radius at 30 m from the light source (Bat Research and Consulting, Tucson, AZ, USA).

Each UV illuminator unit (Figure 2a) consisted of one high-power light-emitting diode (LED) housed with associated circuitry in a heat sink enclosure that dissipated waste heat produced by the LED. The LED’s ultraviolet output was focused by a lens and then passed through a dichroic filter that removed most of the visible components of the LED’s output (wavelengths of light > 400 nm). A switch-mode regulator provided input power to the LEDs as a DC voltage. We modulated output of the power regulator with a pseudo-random pulse generator. Illuminators always operated during this study in a flickering state made possible through a pulse generator that modulated the circuit power regulator, causing the UV light output to switch on and off in a pseudo-random pattern with a frequency of about 0.5 s and a varying duty cycle from about 10% to 90%.

We mounted the 12 illuminator units deployed per turbine 20 m above the ground on the steel monopoles with 15 cm aluminum posts and rubberized magnetic bases (Figure 2a–d; Model NADR257M, Master Magnetics, Castle Rock, CO, USA). The horizontal aluminum posts were fitted with plastic, upward-oriented, white, non-fluorescing bird spikes (Model 3, Bird-B-Gone, Inc., Santa Ana, CA, USA) to ensure they were not used by perching birds. The illuminators were arranged radially and evenly spaced around the circumference of the monopole, with their illumination faces pointing up and tilted −5° off horizontal, away from the monopole (Figure 2a–d). This configuration was intended to cast dim UV light with an illumination intensity of <100 µW/cm^2^ at the turbine surfaces. By angling the beams outward from the monopole, we mostly avoided illuminating the turbine too close to the light source and causing bright spots, rather than the intended dim illumination over broad areas of the upper monopole, nacelle, and much of the rotor-swept zone not shaded from below by the nacelle. While it may have been preferable to mount additional illuminators on the upper nacelle to illuminate shaded areas of the upper rotor-swept zone, the configuration we eventually chose for this study was a compromise between ease of access and complete UV coverage of the turbine surfaces.

We cast dim UV light on the turbine surfaces rather than brighter light to reduce the chances of attracting insects from further distances and causing other unwanted environmental effects. Artificial lights pointed toward animals also have real potential to do harm. Intense UV light shone out from turbines could adversely affect bats and other types of animals, such as night-migrating insects and birds, which wind turbines might otherwise not affect. Bright lights can permanently damage the eyes of dark-adapted animals and light of even moderate intensity can temporarily blind animals with dark-adapted vision, [49] such as bats and nocturnally migrating birds (the latter of which can also see bright UV light [50,51,66]). The effects of intense light on the eyes of nocturnally active animals are poorly characterized, but flying blind at night, even for brief periods, could have catastrophic consequences for bats and birds relying on vision to safely navigate around moving wind turbines. The issue is difficult to address in the context of artificial lighting at night because there are large differences in the eye sensitivity and potential for harm among nocturnal animals [49,50,66]. Therefore, we erred on the side of caution with our UV lighting cue, both in terms of the illumination intensity and the treating of the turbine surfaces rather than directly targeting approaching animals with beams of light.

We programmed timers controlling the power of UV illuminators on each turbine to turn them on every other night throughout the study period, beginning at local civil sunset time and ending at the time of local sunrise, as determined by the timer software. The UV light treatments were staggered between the two turbines, such that nights when one of the turbines was lit with UV, the other was not. Therefore, the wind turbines were our intended experimental unit (but see Results section). The parts and supplies for each UV illuminator system cost approximately USD 6500 when they were built in 2018.

We measured the illumination intensity of our lights and confirmed that we properly implemented the UV light treatments, and that the UV light was illuminating large areas of the turbine surfaces with three different types of specialized UV-sensitive spectrophotometers and cameras. We used a portable USB spectrophotometer sensitive to light ranging from 225 to 1000 nm (Model AFBR-S20M2WU, Broadcom Inc., San Jose, CA, USA) to measure light from an array of three illuminators mounted 2 m above the ground and pointed down a dark residential street at night with no moon visible in the sky. We fit the input tube of the spectrophotometer with a dichroic filter identical to the type used for our illuminators, which prevented light at wavelengths > 400 nm from reaching the sensor. Rather than pointing the input tube of the spectrophotometer directly at the illuminators, we pointed it away from the light source and measured the illumination intensity of the UV light reflecting off a neutral gray camera calibration card mounted 10 cm from the spectrophotometer and directly facing the illuminators. We took spectrophotometer readings at distances ranging from 75 to 5 m from the illuminators, at 10 m intervals, as measured with a laser rangefinder (Model ProStaff 1000, Nikon Corporation, Minato, Japan). The software recorded the spectrophotometer measurements as normalized counts that were averaged over a 1 s sampling period, in units of µ/cm^2^/nm; we then baseline-corrected those values with the “normal” function of the manufacturer’s software (Waves). We approximated the relevant illumination intensity of the cast light by integrating the area under the curve for readings between 300 and 500 nm, reported in µ/cm^2^.

We monitored the nightly function of the lights throughout the study period with imagery from a custom-modified digital camera capable of sensing UV light (Figure 3; Model “Full spectrum, full frame, fused silica sensor cover conversion kit,” Kolari Vision, Raritan, NJ, USA; Model EOS Rebel T5, Canon U.S.A., Inc., Long Island, NY, USA) and by automatically taking long-exposure images at a specified time and duration each night of the study (How to Make a Long Term Time-Lapse, https://www.instructables.com/How-to-make-a-long-term-time-lapse/, accessed on 1 December 2021). This camera was positioned about 1 km west of both turbines and centered between them, and it imaged the landscape that included both turbines and the city of Denver in the background. Throughout the study, we recorded a single 30 s to 40 s long-exposure image each night within an hour of midnight. During most of the study, this camera was fitted with a UV pass filter that blocked all visible light (Model BW403, Schneider Kreuznach, Bad Kreuznach, Germany), but we later realized it was passing near-infrared light, which was more prominent than UV light in the nighttime landscapes we were imaging. Therefore, we changed the pass filter during the summer of 2019 to a type that only passed UV wavelengths and not visible or infrared parts of the spectrum (Ultraviolet Bandpass Transmission Lens Filter, Kolari Vision, Raritan, NJ, USA). The latter filter had a 50% transmission peak at 365 nm and transmission of light through the filter tapered off by 340 nm and 380 nm to <25%. Twice, we imaged the illuminated turbines with a prototype high-resolution, high-speed, ultra-sensitive, scientific-grade camera capable of detecting and imaging single photons in the UV spectrum (Figure 3; loaned from Teledyne Photometrics, Tucson, AZ, USA) and fitted with the same filters that blocked visible light.

### 2.3. Video Recording Animal Presence and Activity

We monitored the rotor-swept airspace of both turbines using pairs of surveillance cameras that imaged in the thermal infrared spectrum (~9000–14,000 μm) and were fitted with 19 mm lenses (Axis Q1932-E, Axis Communications, Lund, Sweden). Camera pairs were operated at a 30-frame-per-second sampling rate, 640 × 480 pixel resolution, and recorded digital imagery in the false-color scheme “Ice-and-Fire” provided with the camera firmware. We magnetically mounted camera pairs approximately 2–5 m above ground level on the east sides of the turbine monopoles, with 1 m spacing between the paired cameras, using an industrial camera mounting base (RigMount X6 Magnet Camera Mounting Platform, Rigwheels, Minneapolis, MN, USA). We aimed cameras straight up the monopoles so that the upper two-thirds of the image included the turbine blades, nacelle, and surrounding airspace, whereas the lower third of the image included the monopole. The video scenes encompassed an estimated 0.30 km^3^ of airspace from a field of view 86.6 m wide by 64.2 m high at a range of 160 m. With this configuration, we were able to observe the presence and the behaviors of the insects, bats, and birds on the leeward side of the turbine (the prevailing wind direction was from the west) at various heights above the ground, including around the UV illuminators. The automatic processing algorithm detected flying animals when their body heat triggered more than one pixel, which depended on animal size and distance from the camera. In our experience, visual classification of animals requires that they are imaged at resolutions higher than approximately 8 pixels/m. We estimated maximum identification distances for this study of 20 m for medium-size insects, such as moths and beetles (~3.5 cm wingspan), 40 m for larger insects (~7.0 cm wingspan), 125 m for bats and similarly sized passerine birds (~20 cm wingspan), and about 150 m for the largest bats and large birds (e.g., ≥25 cm wingspan). Insects were visually discernable from bats and birds by differences in the clarity of image edges, characteristic shapes, and the higher speeds at which they transited the video scenes. Bats and birds were visually discriminated by shape, wingbeat consistency, and flocking; animals observed soaring, flying with others in symmetrical formation, or showing a flap-glide flight cycle were assumed to be birds. We classified distant animals determined not to be insects (e.g., sharp image edges and transiting the video scene slowly over numerous video frames), but which were imaged at insufficient pixel densities for reliable identification, as “high flyers”.

Each camera pair was controlled and powered over the Ethernet through direct-bury CAT5e data cables routed through a power-over-Ethernet network switch (Model GS108PEv, Netgear Inc., San Jose, CA, USA) to a laptop computer (Model 5420, Dell, Round Rock, TX, USA) inside the turbine base (Figure 4). We synchronized the camera and laptop clocks to the second by attaching a Network Time Protocol device to the network switch, which acquired accurate times with a global-positioning-system antenna and receiver (Model TM1000A, Time Machines Inc., Lincoln, NE, USA). The cameras streamed video imagery to the laptop hard drive using software available from the camera manufacturer, where it was temporarily stored (Axis Camera Station 5, Axis Communications, Lund, Sweden). Video recording began at least 30 min before sunset and ran until at least 30 min after sunrise throughout the study period. After the recording ended each morning, we sent video buffered to the hard drive of the laptop in Advanced Systems Format (.asf) to a portable hard drive using the “scheduled export” function of the camera software. We connected a portable hard drive to the laptop with a 4.5 m cable and housed each hard drive in a weatherproof case (Model 1060, Pelican Products, Inc., Torrance, CA, USA) placed outside the turbine tower, where it could be accessed without having to unlock and enter the turbine structure. These video methods were nearly identical to those reported by Goldenberg et al. [65]. Although we recorded synchronized, paired videos from each turbine throughout the study period, we analyzed only video from the left camera of each pair for the analysis reported herein. We will use these paired videos in future efforts to develop computer vision algorithms and processing routines for estimating animal flight height at wind turbines.

### 2.4. Video Processing

We automatically processed thermal video recordings using computer software to detect bats, birds, and insects while ignoring other moving objects such as the wind turbine blades and clouds. Methods of video processing followed those of Cryan et al. [43] and Goldenberg et al. [65]. In short, we converted videos from .asf to frame-indexed Audio Video Interleave format (.avi) and then processed them using previously published code and proprietary software ([43]; MATLAB and Image Processing Toolbox, MathWorks, Natick, MA, USA). This processing routine identifies individual frames in the night-long video where the algorithms detected moving objects (determined through frame differencing) with salient image features, such as bats, birds, and insects. Each frame with an algorithm-detected object was then visually reviewed by a trained video technician who manually assigned it to an object class (bat, bird, insect, high-flyer [defined as an animal flying high enough in the video scene to be visually indistinguishable as a bird or bat], airplane, turbine blade, cloud, or unknown false positive) and recorded the time, duration, number of targets observed, and any behaviors or interactions with the turbine noticed during the video detection. Unlike the past studies that these video processing methods are based on, we added the additional step of counting the number of bird and insect detections, as well as the previously tallied bat counts. Consecutive detections in which a bat went out of camera view but reappeared within 1 min or less were not counted as independent detections and instead were counted as the same event (consistent with previous work by Cryan et al. [43] and Goldenberg et al. [65]). We summed duration (in seconds) of bat flight activity for all the detections within a night. We specifically tallied flight trajectories that indicated potentially high-risk behaviors, including flight within the rotor-swept zone, displacement, and possible strikes. We defined displacement as any event during which a bat was visibly moved through the air after it passed within a few meters of a moving turbine blade but during which there was no visible contact between the bat and the turbine blade. Unambiguous contact between flying animals and moving turbine blades is difficult to determine in thermal imagery. Because we did not conduct concurrent ground searches for bat fatalities around the wind turbines, the events in which a moving blade appeared to make physical contact with a bat are hereafter referred to as “possible strikes”. For comparability, tallies of detections were restricted to the nighttime period delimited by sunset and sunrise. Video reviewers made identifications without knowledge of the nightly treatment-control assignment of UV illumination.

### 2.5. Statistical Analysis

The resulting data acquired in 2019 and available for analysis encompassed the beginning of bird (1 March) and bat (6 April) passage/activity in the study area up until the termination of the UV trial due to technical issues (18 September). Due to an interruption of the UV illumination from 3 July to 11 August, we split analysis of the UV illumination effects into “spring” (before 3 July) and “autumn” (after 11 August) periods. We evaluated the responses of bats, birds, and insects to UV illumination within a single-case experimental design (SCED) framework. SCEDs are widely used in various domains of science, including health and behavioral studies [67], and have been developed for use when only one or a few subjects are available (e.g., individual, school, or town) and involve the deliberate manipulation of an independent variable (in this study, UV illumination). We considered the SCED approach appropriate for this study given that time-series data were only available from a single wind turbine, as detailed in the following results section. The specific SCED used in this study entailed an alternating treatment design that allowed for a rapid and readily reversible treatment/control assignment and comparison of the resulting response [68].

Time-series data often possess an autocorrelated (serially dependent) error structure in which each observation in the series may be correlated to some extent with the preceding observations, and standard methods of statistical analysis (e.g., *t* tests, analysis of variance, and regression) applied to single-case time-series data with autocorrelation can produce inflated Type I error probabilities [69]. However, Levin et al. [69] demonstrated that whenever first-order autocorrelation is moderate (≤0.50) and positive, the Type 1 error rates for an alternating treatment design are less than or equal to the nominal value of 0.05 and attain adequate statistical power (>0.75); this particularly holds for a systematically alternating design based on a single observation per phase (e.g., nightly rotation of control and UV treatment assignment). In all the cases observed in this study, the first-order autocorrelation for each data series was negligible and non-significant or was significantly positive but less than 0.50.

We assessed the effects of the UV treatment on bat, bird, high-flyer, and insect detections relative to the control in the SCED framework with the ALIV method (an abbreviation of “actual and linearly interpolated values” [68,70]). By necessity, our study was a single-case experiment (only the data from one turbine were available) from which control-treatment responses were repeatedly observed. As such, analysis of the autocorrelated observations was not suitable with standard parametric techniques. The ALIV values used in a comparison include both the actual observed values and the linearly interpolated values (i.e., points representing possible values had the condition taken place during a measurement occasion in which the alternate condition was applied). The statistical significance of the ALIV tests was estimated by a Monte Carlo random selection of all possible treatment-control ordering [68]. The ALIV method and randomization test that we applied is a nonparametric technique that makes no distributional assumptions [71]. The method requires stipulating a directional hypothesis (i.e., the effect increases versus the effect decreases activity and the resulting detections). However, considering the exploratory nature of the study, the treatment effects were not anticipated or known a priori to manifest in only one direction. Consequently, we conducted a one-tailed test in each direction in each season and adjusted the *p*-values with the Benjamini–Hochberg (BH) procedure for multiple comparisons to control for family-wise error rate [72], as determined by the number of groups within the taxa. Adjustments were made separately within each taxon (resulting in 12 tests for bats, 4 tests for insects, and 8 tests for birds and high-flyers). ALIV analyses were carried out in R (version 3.5.1; [73]), with code provided in Manolov and Onghena ([68]; the ALIV method is also available as a web application at https://manolov.shinyapps.io/ATDesign/, accessed on 1 December 2021).

## 3. Results

### 3.1. Lighting System Performance

The results demonstrated the feasibility of UV-illuminating turbines and the concurrent monitoring of animal responses to such experimental treatment. The UV illumination was invisible to humans; yet, we confirmed that the turbine surfaces were lit to the highest reaches of the blades (Figure 5). Appendix A summarizes the illumination intensity of three lights at various distances measured by spectrophotometer during off-turbine testing. We installed UV light systems on both wind turbines during the week of 15 August 2018 and they remained installed through October 2019, except for a period from 3 July through 11 August 2019 when we took them down for repair and maintenance. Lighting systems were installed on the turbines for a total of 261 nights and taken down for repair for 41 nights, or 15% of the intended installation period. Multiple prototype illuminator units failed after water incursion or operator error during key seasons of bat activity, but we engineered and implemented fixes during the study and were able to keep the experiment running. The thermal surveillance cameras that monitored the UV lights and surrounding airspace functioned properly and did not require maintenance over the span of 261 nights, with each camera recording more than 3000 h of nighttime imagery during the study. Video files from 6 nights (2%) of the monitoring were lost due to storage equipment failure and video recordings from 19 nights (7%) were lost due to user error during processing. To satisfy the requirements of statistical power and the continuity of data and because the observations were limited when the turbines were taken offline for maintenance, repair, and other experiments, we hereafter report results from only one of the two turbines (North Turbine) and the observations there, spanning from March through September 2019.

Although the UV illuminators experienced failures due to water incursion and power management, we easily fixed those problems, and the lighting generally functioned with little maintenance as specified across all seasons, including winter (Figure 1b). Facility technicians installed each system on a turbine in approximately 4 h using a boom truck that could reach a 20 m height (Figure 2c), and each was installed and removed three times over the course of the study to perform repairs.

### 3.2. Response by Bats, Insects, and Birds

The magnitude and direction of the counts among the taxa relative to the UV treatment were mixed. However, none of the tests was statistically significant, due in part to the multiplicity of tests and the short time series available for each period (Table 1). The number of nightly bat detections markedly increased from spring though summer and into the autumn period (Figure 6 and Appendix A). The UV illumination during the spring period may have increased bat detections by about one to two events per night (ALIV mean difference = 1.6; BH *p*-value = 0.144), and the cumulative duration of detections on nights with UV treatment during spring may have also slightly extended the amount of time (~10 s) that bats were present (ALIV mean difference = 10.0; BH *p*-value = 0.294). Likewise, the count and duration of bat detections during the autumn period trended positive with the UV treatment, but the observed differences were not significant (counts: ALIV mean difference = 6.7; BH *p*-value = 0.372; duration: ALIV mean difference = 66.7; BH *p*-value = 0.486).

The nightly counts of bat detections were highly correlated with the total nightly duration of the events (r = 0.90 in each of both periods), and the pattern of ALIV metrics over time were also quite similar. The correlation indicated that the duration may have been largely driven by the number of nightly detections and that an increase in detection rate as a response to the UV illumination was not compensated by a decrease in event duration.

As with the bat detections, insect activity may have slightly increased during the spring period in response to the UV treatment, although not significantly (ALIV mean difference = 143.9; BH *p*-value = 0.064; Figure 6 and Appendix A). Mean insect numbers during the autumn period were two- to three-fold higher relative to the spring period but also did not exhibit a significant response to the UV treatment (ALIV mean difference = 76.0; BH *p*-value = 0.562). The bird and high-flyer counts were not significantly different between the control and the UV treatments (Figure 6 and Appendix A). Both groups demonstrated relatively similar counts between periods with a low overall rate of nightly detections during mid-summer, indicating that the high-flyer counts were probably comprised of birds rather than bats (Table 1).

The incidence of high-risk bat behaviors observed (that is, flights within the rotor-swept zone, displacements, or possible strikes) was relatively low during the spring period (total = 7) but increased almost twenty-fold over the autumn period (total = 126). Notably, the differences in high-risk bat counts were small and detections were not demonstrably related to the UV treatment in either period (spring period ALIV mean difference = −0.1; autumn period ALIV mean difference = 0.4).

## 4. Discussion

Bats in our study may have responded to the experimental UV illumination of the turbine surfaces, but in a way more suggestive of attraction rather than the intended deterrence. Our results illustrate the challenges of finding device-based methods for predictably manipulating the behaviors of wild bats flying near wind turbines at night. Efforts to develop devices for wind turbines that reduce or prevent bat fatalities are intensifying; yet, no well-proven solution free of incidental environmental effects has emerged to supplant the need for curtailment when unacceptable fatality rates occur. Because fatality reduction devices for bats at wind turbines constitute an incipient technological field, scientists and engineers developing and testing such systems face the difficulty of designing systems that work but that do not cause additional problems in the process.

A known risk when initiating this research was the possibility of attracting insects to the UV lights used for dimly illuminating the turbine surfaces. Insects physiologically sense extremely dim UV light and are attracted to certain types of artificial light at night (ALAN; [74,75]). However, the details of how insect visual systems sum dim light inputs to inform behavior and whether pure UV light absent of other spectral components is as attractive as multi-spectral light sources remain open areas of research [74,75,76]. For example, LED lights may not attract insects as readily as other types of lights, and cold white LEDs might deter insects [77]. Flickering light also seems to be less attractive to insects than steady sources of illumination [78], which was why we designed our turbine UV lights to continuously flicker at a high duty cycle. Entomologists use UV LED lights for insect attraction studies, but those lights typically have very different properties to what we designed and include intense peaks in the visible parts of the spectrum [76]. When planning this study, we found little evidence to suggest that insects would be attracted to our dim, flickering UV lights, filtered of visible components (>400 nm) at distances greater than approximately 50 m. In a prior study that used nearly identical UV illuminators with visible-light filters, UV treatment of forest habitat caused a small statistically significant increase in insect activity, albeit one not associated with a concurrent increase in bat activity [63]. Notably, in the current study we did not detect significant increases in insect activity associated with the UV treatment of the turbine over the course a year, indicating that bat attraction to UV-attracted insects may not be a concern with this technology. Much remains to be learned about how insect behaviors change around different kinds of light sources, and new and accessible tools are available for scientists interested in such questions [79].

As with insects, predicting the influence of ALAN on bat behaviors remains a challenge. Research into how bats in natural settings respond to ALAN has intensified in recent years as concerns grow about the “light pollution” of natural habitats. Among bats, species composing regional communities often respond in different ways, with both spectral composition and intensity influencing response; few patterns of bat response to ALAN apply to all species [80,81]. In general, experiments consistently demonstrate certain species of bats avoiding experimentally lit areas of landscapes, whereas other species in a local bat community might forage opportunistically on insects at the lights [82,83]. For example, experimental treatment with ALAN reduced drinking by forest bats in Italy [84], and in an urban area of Germany, greater noctule bats (*Nyctalus noctula*) tended to avoid lit areas but opportunistically foraged in some situations [85]. In other urban settings, the activity of some species was reduced around LEDs compared with mercury vapor lamps, whereas other light-averse species were more likely to forage around LEDs, and experimental evidence indicated it was spectrum rather than illumination intensity influencing these differences [86]. As was experimentally demonstrated with cowbirds, the spectral composition of light, including chromatic and achromatic contrast, plays an important role in how some vertebrates perceive and behaviorally respond to artificial light [87]. Artificial illumination in simulated caves was shown to decrease the activity of bats, with red light having less of an effect on four species (three families, two suborders) of European bats (*Myotis capaccinii*, *Miniopterus schreibersii*, *Rhinolophus mehelyi*, *R. euryale*; [88]). Further demonstrating the different responses of bats to lights with different spectral properties, certain agile-flying species of bats in the Netherlands avoided white and green light in experimentally lit landscapes, whereas others occurred more around these areas, and most bats showed no clear response to landscapes lit with red light [89].

Much of the variability in how bat activity changes with the intensity (brightness) and spectral composition (color) properties of artificial light sources in experiments may be difficult to separate from the influence of insect prey driving bat presence. Bats might take advantage of novel predation opportunities in anthropogenic landscapes, which can give them an advantage over less opportunistic species [90]. Two species of pipistrelle bats (*Pipistrellus nathusii* and *P. pygmaeus*) migrating along a shoreline of the Baltic Sea in late summer seemed attracted to experimental green lights on poles, but this attraction was not necessarily associated with foraging [91]. In another subsequent experiment at the same site, bats responded with positive phototaxis to experimental red, but not white, lights, yet fed at the white lights [92]. Because we observed slight but not statistically significant increases in both bats and insects during the nights that we treated wind turbine surfaces with dim UV light in this study, we cannot rule out the possibility that the bats were attracted to insects and not necessarily to the UV light. For example, the pattern we observed could result if bats investigating the turbine at night were more likely to remain in the area to feed on insects attracted to the light. However, in our analysis of the thousands of hours of thermal video gathered for this study, we saw no instances of bats making repeated prey attacks in flights that focused on the airspace immediately surrounding the UV lights. Furthermore, we regularly observed moths and other large insects resting and crawling on the turbine monopole surfaces between the cameras and illuminators, with and without UV treatment, but never saw bats glean or closely approach such insects. We conclude that there was no causal association between the slight increases we observed in insect and bat activity during UV treatment, but see below for how this relationship could be explored in future studies.

Some researchers have speculated that red lights deployed for aviation safety might influence bat presence near wind turbines, although evidence is limited [3,42,43,93]. Fatality rates of eastern red bats (*Lasiurus borealis*) at turbines in Texas lit with red flashing aviation lights were lower than at unlit turbines [93], although red light has spectral characteristics that are likely to place it near the margin of bat visual perception [52]. Earlier studies led to the speculation that bats might be able to see tall structures from very far distances, that bats might see UV light reflecting off turbines at night, and that bats may visually respond to artificially illuminated structures [3,39,43]. Only one previous study reported manipulating the behaviors of bats in the wild using dim, flickering UV light [63]. In that work, Gorresen et al. [63] used thermal-imaging cameras and acoustic detectors to find that UV treatment decreased bat vocalization rates, and they inferred that increases in the duration of detections were caused by bats flying farther from the cameras and thus remaining in the field of view longer (i.e., “pushed” out of the illuminated area). Bats probably adjust echolocation and vision appropriately for sensing as environmental light levels and other circumstances change [94,95]. Social interactions might also influence this sensory balance during the time of year when bat fatalities at wind turbines tend to peak [63,96,97]. The proportional reliance of bats on sound- and light-based cues as they approach wind turbines at night might be relevant to developing effective solutions for reducing fatality.

We consider the UV light treatment used in this study an indirect cue to approaching bats, where the perceived cue is the illuminated surface of the wind turbine rather than the more intense illumination interface of the lighting device itself. We were able to image the turbines treated with UV light in our study from several kilometers away at night (Figure 5c,d, Appendix A). Spectral measurement and landscape imaging of the UV light we cast across the turbine surfaces confirmed that it was dim (low illumination intensity), as intended (Figure 5, Appendix A). In addition to theoretically supplementing subtle sensory cues already available to the bat’s decision-making processes in its natural environment (rather than a focus on triggering a startle response), the treatment was not visible to human observers or presumably other vertebrate animals known to respond to visible sources of light at night, particularly birds. Brighter UV light shone at night on line markers to enhance their visibility along a stretch of power line in Nebraska substantially reduced the collisions and fatalities of migrating sandhill cranes (*Antigone canadensis*; [28]). Although the eyes of some nocturnal birds, such as owls and certain species in the family Caprimulgidae (e.g., nighthawks, nightjars, and poorwills), approach or equal the light sensitivity of bats, they are not known to possess vision in certain parts of the spectra where bats can see at low illumination intensities, such as UV [50,98,99,100,101,102,103]. Compared with bats, birds tend to have much lower spatial contrast sensitivity and differentiate dimly lit objects from the visual noise of dark backgrounds [48,104,105]. The eyes of bats also trend toward lower refractive power than birds, which determines their ability to gather light across a scene and/or a wide field of view [36,106]. Although many birds see UV light, most are primarily active during the daytime when UV intensity is very high and extreme sensitivity to that part of the spectrum could cause retinal damage [101,107]. The activity of birds migrating past an island in spring and monitored with an avian radar system decreased when high-intensity UV and violet lights were experimentally cast into the airspace [108]. Our initial concern that the brighter illumination surfaces of the upward-pointing UV lights in this study might have been visible and attractive to passing birds was not realized. Our results indicate that birds were not attracted to or deterred by UV treatment of the wind turbine surfaces at night.

Although bat and insect activity may have slightly increased with UV treatment and our low sample size resulted in an inability to statistically detect such change, we saw no indication of increased risky bat behaviors or strikes. If UV light treatment did indeed subtly influence the bat and insect presence, it did not measurably change risk. Remaining uncertainty about the reasons why bats come into close contact with wind turbines seems sufficient justification for proceeding cautiously when attempting to manipulate bat behaviors in relation to wind energy. Overall, we are confident that our indirect, low-intensity UV lighting method minimized risk to non-target animals, and that bats or birds were not unnecessarily harassed, blinded, or injured by the stimulus cue. Our findings illustrate the value of exercising caution with emerging technologies intended to influence animal behaviors.

We uncovered subtleties in how bats might respond to potential fatality reduction methods in this first test of an experimental UV system, which in turn relates to how we measure the efficacy of such methods. To date, the most common metric of efficacy for fatality reduction tests has been to compare the number of dead bats found beneath wind turbines following nights with and without applied treatments (e.g., curtailment, acoustic deterrents [20,21,25,26,27,109,110]). This gold standard of counting carcasses to judge if a method works is fully justified and supported by robust statistical measures and laboratory techniques to enhance the quality, precision, and reliability of such data [111,112,113,114,115,116,117,118,119,120,121]. Had we relied on fatality ground searches as a metric of bat response to dim-UV treatment, we would not have been able to statistically test for treatment effects, by either attraction or deterrence, simply (and fortunately) due to the low sample size. While we do not advocate settling for less than physical evidence of fatality when testing new technologies, we do see value in gathering more subtle measures of animal response to experimental treatments.

## 5. Conclusions

Our experience in this study demonstrates how the positive or negative effects of fatality reduction attempts may be subtle, and efficacy metrics such as fatality searches might not detect them. Thermal-imaging cameras have long played an important role in discovering and characterizing the behaviors and seasonal activity patterns of bats around wind turbines [15,43,65,122,123,124,125,126]; the current study was no exception. Thermal imaging let us quantify bat and insect presence and activity levels near the experimentally UV-treated turbines, as well as observe the types of behaviors they exhibited in response. Observations of bats with thermal-imaging cameras reveal behaviors that could be associated with risk and therefore quantified as an efficacy metric when applied to the bat turbine interaction problem. Such behaviors include leading–following behavior, near strikes, displacements (“rotor-disrupted flights”), possible and likely strikes, hovering, blade chasing, and dodging through sweeping rotors [122,124,126,127]. Fatality searches cannot reveal risk at the degree of temporal precision possible with thermal imaging, but the two can be associated [126]. An additional benefit of using thermal-imaging cameras to measure more subtle responses than fatalities is that it opens the research to a wider variety of situations and facilities. For example, we observed on average only a few bats per night and very few possible strikes in the 18 months of video we analyzed from this site; yet, we gathered meaningful information relevant to assessing the effects of dim UV illumination on animal presence and behaviors.

We were not sure whether UV illumination might have increased bat activity through attraction to UV illumination independently of possible increased insect presence, but we might gather additional measures in future experiments to answer such questions. Because many of the medium-size insects (<3.5 cm wingspan) were detectable only up to about the height of where we mounted the lights on the turbine monopole (20 m) and larger insects (<7.0 cm wingspan) only about halfway up the monopole (40 m), the general detection area of insects was disproportionately lower to the ground than the larger detection area of bats. We did not try to estimate the heights of bats or sample for insects near the highest reaches of the turbines where bats were discernable (125–150 m). We therefore cannot assess whether the activity of insects and bats might have increased consistently near all parts of the turbine that were illuminated or just at the lower reaches of the monopole near the lights. For example, could a preponderance of near-ground activity by local bats visiting familiar foraging grounds have caused the possible increased activity we saw with UV treatment out of the higher risk areas around the moving blades? In addition, we did not record the timing of insect detections, which could have provided insights into whether insects and bats were more likely to co-occur at turbines lit with UV light.

We based our conclusions on observations gathered only at a single wind turbine that serves primarily as an energy research facility. The low sample size, equipment failures leading to downtime of the experimental treatments, and the unique context of the setting should not be dismissed. By necessity, we relied on a single-case analysis framework, which focuses on the internal validity of a study, where the objective is to make inferences about a treatment’s effect in a specific experiment with a specific subject [128]. As such, inference from this type of test to other settings (i.e., external validity) is not suitable unless systematic replications of the single-case experiments are conducted [129]. Despite evidence that the bats might have been slightly attracted to the dim UV illumination during this experiment, the results do not demonstrate an associated change in fatality risk. This suggests that further, yet cautious, exploration of light-based systems might be reasonable.

## 6. Patents

P.M.C. and P.M.G. co-authored a United States Patent (9995282B2; Cryan et al., “Selectively Perceptible Wind Turbine System”, Pub. Date: 12 June 2018) on behalf of the U.S. Department of Interior and the University of Hawaii at Hilo.

## Figures and Tables

**Figure 1 animals-12-00009-f001:**
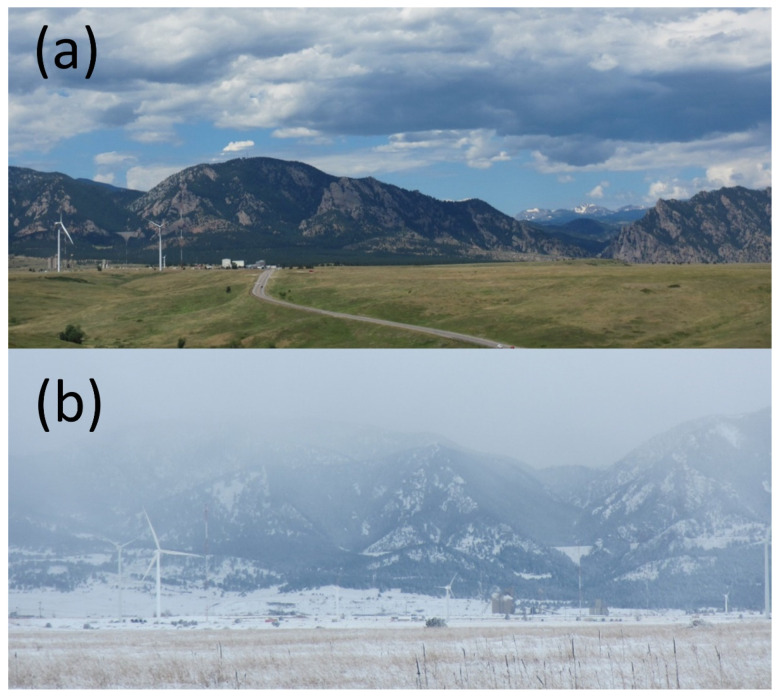
Wind turbines at the National Wind Technology Center, National Renewable Energy Laboratory, near Boulder, CO, USA. Ultraviolet (UV) illuminator systems to test the influence of dim UV illumination on flying animal presence and activity were installed during summer 2018 (**a**) and left in place and operating through the winter of 2018/2019 (**b**). Images by Paul Cryan.

**Figure 2 animals-12-00009-f002:**
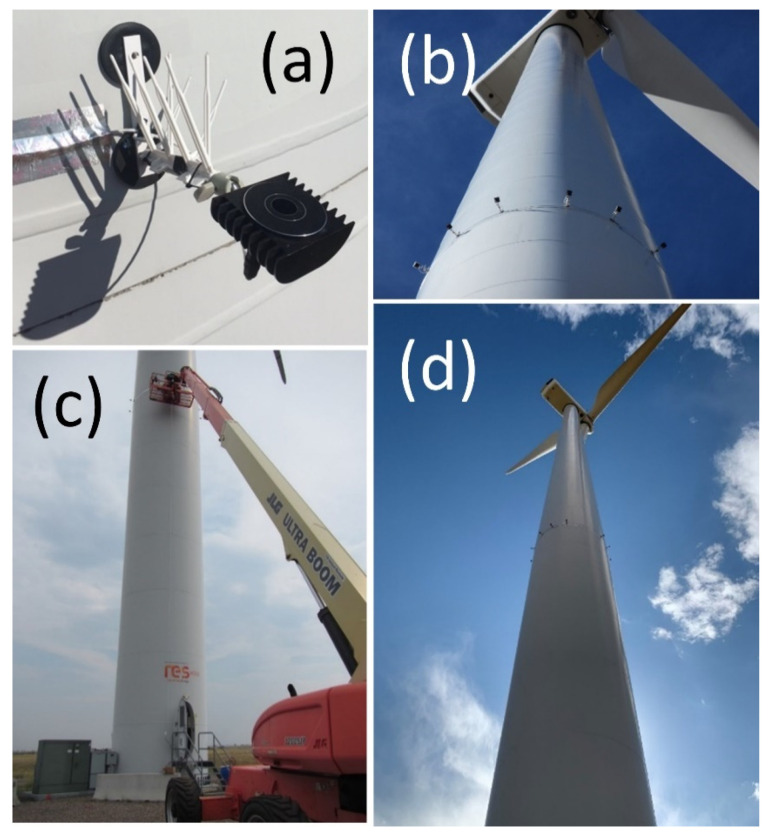
UV illuminator units (**a**) magnetically attached around the circumference of wind turbine tower in an array of 12 (**b**) and installed 20 m above the ground with a boom truck (**c**). Light arrays were positioned below the sweep of the wind turbine blades and one-quarter of the way up the turbine monopole (**d**) so they lit the blades, upper monopole, and underside of the nacelle. Images by Paul Cryan.

**Figure 3 animals-12-00009-f003:**
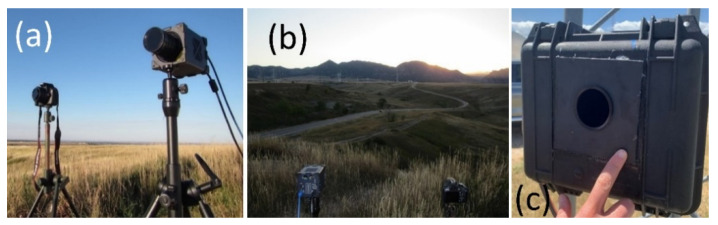
Two types of specialized cameras were used to periodically image turbines from afar and confirm and visualize dim UV illumination of wind turbine surfaces (**a**,**b**). Time-lapse, long-exposure landscape camera captured nightly images of the UV-illuminated turbines throughout the study period (**c**). Images by Paul Cryan.

**Figure 4 animals-12-00009-f004:**
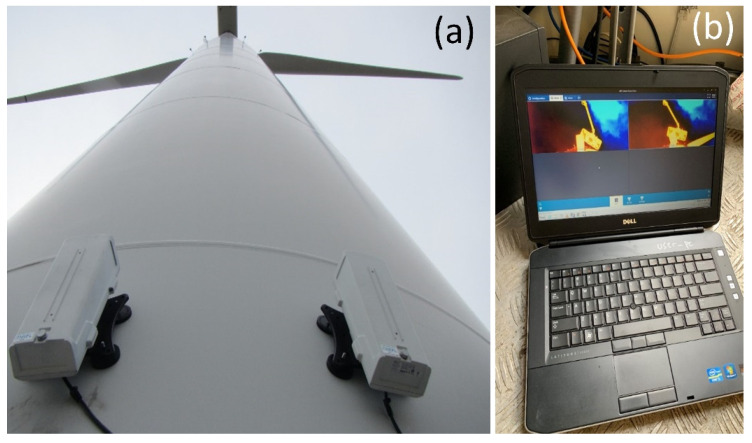
Bat, bird, and insect activity was monitored with pairs of time-synchronized thermal-imaging surveillance cameras mounted on the wind turbines (**a**). Imagery was automatically recorded to a laptop computer inside the base of the turbine (**b**) each night before being exported to a portable hard drive outside the turbine each day. Images by Paul Cryan.

**Figure 5 animals-12-00009-f005:**
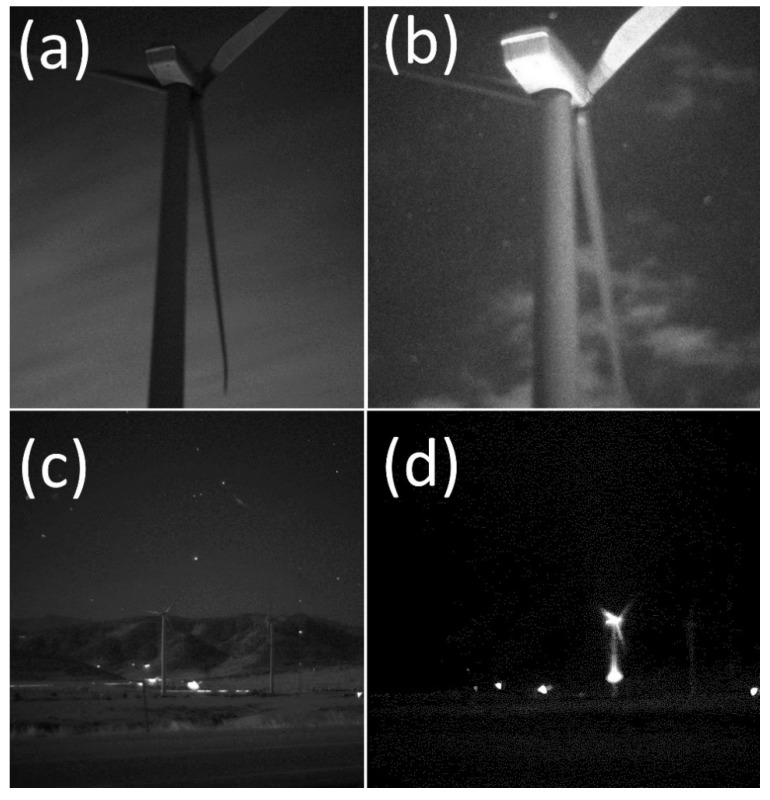
Nighttime images of wind turbines imaged with UV-sensitive cameras. Images of wind turbine captured from 60 m away with a high-sensitivity, scientific-grade camera without UV illumination (**a**) and with illuminator system turned on (**b**). Panels (**a**–**c**) show nighttime images through a light filter that blocked visible parts of the spectrum (~400–700 nm) but allowed both UV and near-infrared wavelengths (~700–800 nm) to pass. Bottom row shows turbines imaged from approximately 1.5 km to the northeast with both turbines with UV illumination and through the filter that allowed passage of UV and near-infrared wavelengths (**c**), and the same scene with the North Turbine (right) UV-illuminated and through a filter that allowed only UV wavelengths to pass and blocked all infrared and visible wavelengths (**d**). Images by Paul Cryan.

**Figure 6 animals-12-00009-f006:**
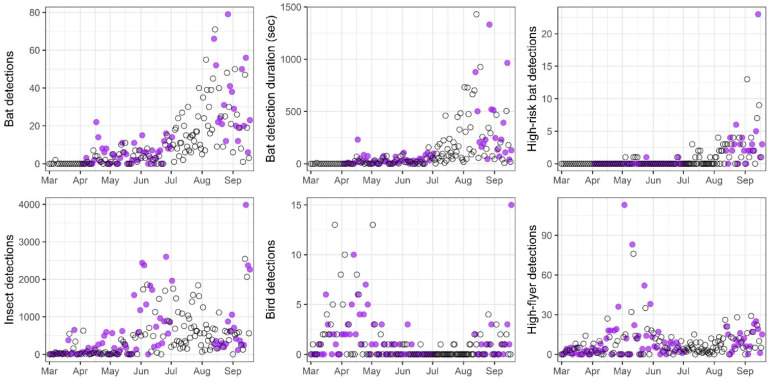
Response to UV illumination on nights with (purple circles) and without UV (open circles) treatment based on counts of bats (all detections, cumulative duration of detections, and high-risk detections), insects, birds, and “high-flyers”.

**Table 1 animals-12-00009-t001:** Descriptive statistics of nightly detection counts for each group (“bats”, “insects”, “birds”, and “high-flyers”) by test period (“spring” and “autumn”) and control (UV = 0) and treatment (UV = 1) nights. The group “bats-duration” is the total nightly duration (seconds) during which bats were detected. The group “bats-high-risk” is the total nightly count of detections in which bat flight occurs within the rotor-swept zone. “Total” is the sum of detections for all control and treatment nights. Additional metrics include the mean difference between actual and linearly interpolated values (ALIV) of control and treatment, and *p*-values for each directional test of treatment effect (Reduce, Increase) as adjusted by the Benjamini–Hochberg (BH; [72]) procedure separately for each taxon.

Group	Period	UV	Nights	Total	Mean	SD	Median	Min	Max	ALIV Mean Difference	Reduce BH *p*-Value	Increase BH *p*-Value
bats	spring	0	44	157	3.5	3.8	2.5	0	15	1.6	0.981	0.144
		1	44	234	5.3	5.3	4.5	0	22			
bats	autumn	0	19	502	26.4	18.8	25.0	1	71	6.7	0.981	0.372
		1	19	621	32.7	19.3	24.0	6	79			
bats—duration	spring	0	44	1046	23.8	32.1	14.5	0	156	10.0	0.981	0.294
		1	44	1538	35.0	42.9	24.0	0	230			
bats—duration	autumn	0	19	5899	310.5	358.1	178.0	6	1430	66.7	0.486	0.486
		1	19	6884	362.3	343.5	229.0	39	1331			
bats—high-risk	spring	0	44	5	0.1	0.3	0.0	0	1	−0.1	0.372	0.981
		1	44	2	0.1	0.2	0.0	0	1			
bats—high-risk	autumn	0	19	59	3.1	3.3	3.0	0	13	0.4	0.981	0.567
		1	19	67	3.5	4.9	2.0	0	23			
insects	spring	0	62	17,627	284.3	456.0	44.5	0	1856	143.9	0.986	0.064
		1	62	27,829	448.9	668.8	145.0	0	2601			
insects	autumn	0	19	12,601	663.2	621.0	568.0	0	2545	76.0	0.947	0.562
		1	19	15,080	793.7	983.6	354.0	113	3987			
birds	spring	0	62	103	1.7	3.0	0.0	0	13	−0.3	0.767	0.767
		1	62	86	1.4	2.1	0.0	0	10			
birds	autumn	0	19	20	1.1	1.2	1.0	0	4	−0.1	0.767	0.767
		1	19	29	1.5	3.3	1.0	0	15			
high-flyers	spring	0	62	483	7.8	11.7	4.0	0	76	2.5	0.895	0.712
		1	62	634	10.2	19.0	4.0	0	113			
high-flyers	autumn	0	19	238	12.5	9.0	12.0	0	29	−1.2	0.767	0.767
		1	19	212	11.2	7.7	11.0	0	25			

## Data Availability

The data presented in this study will be openly available: Cryan, P.M., Gorresen, P.M., Straw, B.R., Thao, S., and DeGeorge, E., 2021, Bat, insect, and bird activity at a wind turbine in Colorado experimentally illuminated with ultraviolet light at night in 2019 to try and deter bats: U.S. Geological Survey data release, https://doi.org/10.5066/P9M0S3BV, accessed on 4 December 2021.

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
