# Peer review of "Influencing Activity of Bats by Dimly Lighting Wind Turbine Surfaces with Ultraviolet Light"

_animals, 2021, doi:10.3390/ani12010009_

Round 1

Reviewer 1 Report

The authors investigated whether flickering UV light decreases the activity of bats at wind turbines by checking visually whether bats, insects, and birds were repelled (or attracted) to wind turbines illuminated by UV stroboscopic light. The authors focused on a single turbine that they treated alternatively with UV light or as dark control. They observed an increase in bat and insect activity, but were not able to conclude on the negative or positive effects of the UV light treatment. 

The dilemma of this study is the low sample size with only a single turbine investigated. This dilemma is hard to resolve given the huge effort in recording the response behavior of birds, insects, and bats. While I applaud the authors for their effort, it is hard to assess what the outcome of the experiment is, i.e. whether it is generalizable. I am also a bit perplexed how the authors could infer the presence of small objects like insects from distance, and also on the vulnerability of bats without 3D technique. The manuscript is wordy in many parts, and some sections are even redundant. Therefore, the authors should revise the manuscript with the intention of removing any double or triple explanations. Both introduction and discussion could be shortened by at least 1/3. Also, it would be relevant to remove some of the unnecessary information such as the costs for some of the instruments (because anyways this will change over time). I am not an expert on the statistics that the authors have chosen, but see major limitations given the fact that this study is built on n=1. I am further concerned about the inability of differentiating species based on the videos. An acoustic survey would have served better at least for the bat species that were at the core of this paper. Overall, I see some scientific merit in this study, but frankly, I am struggling with where to go from here. A paper should not end with a plea for larger sample sizes or replicates. If it does, it rather looks like a pilot study that could be published as a short communication in a very much condensed way. My suggestion would be to boil this down to a 2000 word short communication that could inform future research efforts. Written as it is, it fails to reach the standards of an original full paper.  

Author Response

Author reconciliation of comments from Reviewer No. 1 of ‘Influencing activity of bats by dimly lighting wind turbine surfaces with ultraviolet light” by Cryan et al.  Author responses in blue.

Reviewer No. 1 Comments

( ) Extensive editing of English language and style required
( ) Moderate English changes required
(x) English language and style are fine/minor spell check required
( ) I don't feel qualified to judge about the English language and style

Yes

Can be improved

Must be improved

Not applicable

Does the introduction provide sufficient background and include all relevant references?

( )

(x)

( )

( )

Is the research design appropriate?

( )

( )

(x)

( )

Are the methods adequately described?

( )

(x)

( )

( )

Are the results clearly presented?

( )

( )

(x)

( )

Are the conclusions supported by the results?

( )

( )

(x)

( )

Comments and Suggestions for Authors

The authors investigated whether flickering UV light decreases the activity of bats at wind turbines by checking visually whether bats, insects, and birds were repelled (or attracted) to wind turbines illuminated by UV stroboscopic light. The authors focused on a single turbine that they treated alternatively with UV light or as dark control.

We chose not to focus any manuscript discussion on the fact that we initially attempted to experimentally treat two turbines with UV light for this study. As described, we ultimately could only use data from one of the turbines and thus fell back from the initial plan to use alternately lit turbines as the experimental unit in a single-case statistical design. As some of the later comments from the reviewer indicate, detailing the challenges of finding a wind facility willing to conduct this type of experiment and the logistics involved in keeping the equipment running seemed of less interest to readers of Animals than the more biological findings on which we focused.

They observed an increase in bat and insect activity, but were not able to conclude on the negative or positive effects of the UV light treatment. 

We thank the reviewer for this observation, which helped us find the inconsistencies in the structure and logic flow of the discussion. In the revised version of the manuscript, we further clarify that UV treatment significantly increased bat and insect activity, but that risk did not seem to increase with activity.  Readers might assume that increased activity of bats or insects close to wind turbines is proportional to risk, but we did not see evidence that that is necessarily the case. In addition to several paragraphs of the results and discussion sections that touch upon these subjects being moved to more appropriate places, we further clarified the first mention of these patterns in the last paragraph of the results section as follows:

The magnitude of seasonal counts and the effects of UV treatment among taxa and sample periods were mixed (Table 1). The number of nightly bat detections markedly increased from spring though summer and into the autumn period (Figure 6 and Figure S2). UV illumination during the spring period significantly increased bat detections by about one to two events per night (ALIV mean difference = 1.6; BH p-value = 0.024). Although not statistically significant, the cumulative nightly duration of detections with UV treatment during spring may have slightly extended the amount of time (~10 s) that bats were present (ALIV mean difference = 10.0; BH p-value = 0.098). Likewise, although count and duration of bat detections during the autumn period trended positive with UV treatment, the observed differences were not significant (counts: ALIV mean difference = 6.7; BH p-value = 0.214; duration: ALIV mean difference = 66.7; BH p-value = 0.243).

The dilemma of this study is the low sample size with only a single turbine investigated. This dilemma is hard to resolve given the huge effort in recording the response behavior of birds, insects, and bats. While I applaud the authors for their effort, it is hard to assess what the outcome of the experiment is, i.e. whether it is generalizable.

We agree that our dilemma was low sample size and that it is hard to assess the scope of inference of our findings. Although we only present data from a single turbine in Colorado, we do demonstrate statistically that we attracted bats and insects to the wind turbine using light that was invisible to humans (and presumably birds). The bat attraction was contrary to our intended response, but it is an interesting result we think is worth reporting in the context of increasing efforts to develop device-based methods of preventing bat fatalities at wind turbines. By first testing a new technique on a small number of turbines such as we did with this research, we limited any adverse effects (unintended consequences) from the experimental treatment, such as if UV light had led to statistical increases in fatalities of bats and birds. Where our inference starts thinning is around the contrary result of not observing sufficient evidence to conclude that the increased bat activity we caused put those attracted bats at risk. That is a puzzling finding that might be an artifact of low statistical power in the study design, height bias in our monitoring methods (which we discuss), or other biological reasons. We hope that the revisions to the manuscript described in the paragraph above, as well as additional changes made in response to other reviewer questions below help convince readers that this was a worthwhile study.

I am also a bit perplexed how the authors could infer the presence of small objects like insects from distance, and also on the vulnerability of bats without 3D technique.

This is a good question that made us realize that, since this is the first study of this kind to report bat, insect, and bird activity around wind turbines with thermal surveillance cameras, that we need to provide more information. The prior studies we reference in the paper, such as Horn et al. 2008, Cryan et al. 2014, and Goldenberg et al. 2021, detail in general how insects, bats, and birds are visually discriminated in thermal video without needing detailed height information from 3D analysis. In short, insects detected by the thermal cameras are flying closer to the cameras so their speed and focal characteristics relative to size and shape are very different than birds and bats flying farther from the cameras. Three-dimensional imaging is helpful for determining the precise 3D positions of flying animals in imagery (on the order of centimeters). However, once an animal is visually classified by a human observer, the known focal length and pixel resolution of the camera, combined with the possible sizes and flight speeds of the types of animals being imaged, makes it possible to visually estimate flight heights (on the order of meters) from a single camera view. We added clarification to the methods section of the revision with text such as,

The automatic processing algorithm detected flying animals when their body heat triggered more than one pixel, which depended on animal size and distance from the camera. In our experience, visual classification of animals requires they are imaged at resolutions higher than approximately 8 pixels/m. We estimated maximum identification distances for this study of 20 m for medium-size insects like moths and beetles (~3.5 cm wingspan), 40 m for larger insects (~7.0 cm wingspan), 125 m for bats and similarly sized passerine birds (~20 cm wingspan), and about 150 m for the largest bats and large birds (e.g., ≥25 cm wingspan). Insects were visually discernable from bats and birds by differences in the clarity of image edges, characteristic shapes, and higher speeds at which they transited the video scenes. Bats and birds were visually discriminated by shape, wingbeat consistency, and flocking; animals observed soaring, flying with others in symmetrical formation, or showing a flap-glide flight cycle were identified as birds.

And in the discussion revised the relevant paragraph to read:

 …Because many of the medium-size insects (<3.5 cm wingspan) were detectable only up to about the height of where we mounted the lights on the turbine monopole (20 m) and larger insects (<7.0 cm wingspan) only about halfway up the monopole (40 m), the general detection area of insects was disproportionately lower to the ground than the larger detection area of bats. We did not try to estimate the heights of bats, or sample for insects near the highest reaches of the turbines where bats were discernable (125-150 m). We therefore cannot…

The manuscript is wordy in many parts, and some sections are even redundant. Therefore, the authors should revise the manuscript with the intention of removing any double or triple explanations. Both introduction and discussion could be shortened by at least 1/3.

We agree with this assessment and apologize for the accidental duplication of text while copying into the journal template. The revision removes the accidental redundancies, unintentional but pedantic-in-hindsight redundancies, as well as irrelevant concepts and tangential information.  We reduced the introduction section by 16% and the discussion section by 26% with the revision. 

Also, it would be relevant to remove some of the unnecessary information such as the costs for some of the instruments (because anyways this will change over time).

We believe the strengths of this manuscript are the new methods we report in detail. Cost of the experimental light systems is among the most frequent questions we are asked about this research thus we chose to include it.  However, to address this reviewer comment we provide additional context by addition of information, “Parts and supplies for each UV illuminator system cost approximately $6,500 USD when they were built in 2018.” We believe this is useful information considering that cost of other systems for bats and turbines can range into the tens of thousands of dollars per wind turbine equipped.

I am not an expert on the statistics that the authors have chosen, but see major limitations given the fact that this study is built on n=1.

Please see detailed responses below to statistical questions from the other reviewer about the study design and single-case studies.

I am further concerned about the inability of differentiating species based on the videos.

As discussed in the answers above, we are confident that our classifications of insects, bats, and birds are legitimate, and that the additional category of high flyers sufficiently accounts for those animals flying so high as to not be reliably classified.

An acoustic survey would have served better at least for the bat species that were at the core of this paper.

We agree that acoustic monitoring might have helped us determine the species of bats being observed, but the logistics of operating acoustic detectors on top of wind turbines, as well as their inconsistent usefulness for detecting bats when present, influenced our decision not to incorporate them into this study. Several of our prior studies support this decision, as bats were more consistently detected close to turbines or UV lit areas using cameras than with acoustic detectors. Furthermore, emerging evidence suggests that hoary bats, which are found dead at wind turbines more than any other, forgo echolocation for behavioral reasons during the time when they are most susceptible to wind turbines.

Gorresen, P.M.; Cryan, P.M.; Dalton, D.C.; Wolf, S.; Johnson, J.A.; Todd, C.M.; Bonaccorso, F.J. Dim ultraviolet light as a means of deterring activity by the Hawaiian hoary bat Lasiurus cinereus semotus. Endangered Species Research 2015, 28, 249-257, doi:10.3354/esr00694.

Gorresen, P.M.; Cryan, P.M.; Huso, M.M.; Hein, C.D.; Schirmacher, M.R.; Johnson, J.A.; Montoya-Aiona, K.M.; Brinck, K.W.; Bonaccorso, F.J. Behavior of the Hawaiian hoary bat (Lasiurus cinereus semotus) at wind turbines and its distribution across the North Ko’olau Mountains, O’ahu. . Hawai‘i Cooperative Studies Unit Technical Report 2015, HCSU-064.

Corcoran, A.J.; Weller, T.J. Inconspicuous echolocation in hoary bats (Lasiurus cinereus). Proceedings of the Royal Society of London B 2018, 285, doi:10.1098/rspb.2018.0441.

Corcoran, A.J.; Weller, T.J.; Hopkins, A.; Yovel, Y. Silence and reduced echolocation during flight are associated with social behaviors in male hoary bats (Lasiurus cinereus). Scientific Reports 2021, 11, 18637, doi:10.1038/s41598-021-97628-2.

Gorresen, P.M.; Cryan, P.M.; Montoya-Aiona, K.; Bonaccorso, F.J. Do you hear what I see? Vocalization relative to visual detection rates of Hawaiian hoary bats (Lasiurus cinereus semotus). Ecology and Evolution 2017, 7, 6669-6679, doi:10.1002/ece3.3196.

Gorresen, P.M.; Cryan, P.M.; Tredinnick, G. Hawaiian hoary bat (Lasiurus cinereus semotus) behavior at wind turbines on Maui. Hawai'i Cooperative Studies Unit Technical Report Series 2020, HCSU-093.

Overall, I see some scientific merit in this study, but frankly, I am struggling with where to go from here. A paper should not end with a plea for larger sample sizes or replicates. If it does, it rather looks like a pilot study that could be published as a short communication in a very much condensed way.

We hope that the revised and condensed version provides a clearer view to the merits of this research. Because of the tremendous effort involved in trying a high-risk experiment of this type and the many novel methods we developed and uniquely applied for the study, we believe it is worth sharing the story of how we dealt with the constraints of applied research.

My suggestion would be to boil this down to a 2000 word short communication that could inform future research efforts. Written as it is, it fails to reach the standards of an original full paper.  

We would prefer to include the detailed methods so that others looking for ways to test deterrence methods and their subtle influence on the behaviors of various types of flying animals can find useful information, regardless of whether they choose to work with UV light. Furthermore, this manuscript currently represents a thorough review of what is known about the spectral sensitivity of bat vision in relation to birds and insects, and in the context of perception of and fatality risk at wind turbines.

(SEE ATTACHED PDF DOCUMENT WITH COPIES OF THESE RESPONSES AND A MARKED COPY OF THE MS WITH CHANGES MADE DURING REVISION)

Reviewer 2 Report

This study assesses the feasibility and efficacy of using dim UV light to deter bats from interacting with wind turbines, providing an important progress report on innovations in fatality reduction measures. The background and methodological description are thorough. An impressive amount of work went into the system design and data analysis. This being a preliminary study, the sample size is very limited. The authors used an appropriate statistical approach in their analysis and are careful to delineate the scope of inference they are able to draw.

Places for improvement in the manuscript include some repeated text and the opportunity for brevity. I would encourage the authors to check their presentation of the results to make sure they are being consistent in their treatment of non-significant results. Some are presented as suggestive of an effect while others are presented as the absence of effect. The criteria for this distinction should be made a bit more clear. It would also be helpful if the methods included some description of the signal provided by birds, bats, and insects during video analysis. What is the rough range of detection of each? What size object (perhaps in pixels?) was detectable by the frame-differencing algorithm used to flag periods of activity? The discussion notes that most insect observations were quite close to the cameras.

Specific comments:

82: The use of triggers for operational changes is repeated from previous sentence.

102: The justification for using light seems overwrought. Bats can see it from far away. Atmosphere and the speed of light? Probably minor points.

109: Can you shorten/combine this and the following paragraph? Also, the topic sentence here really belongs with the previous paragraph.

137: It is unnecessary to label the study as speculative. The hypothesis that UV light could be a deterrent is well-supported and the need for new deterrent methods is convincing.

157: Structure this sentence to highlight effect on bats.

218: This and the next sentence include some repetition of information.

340: This is a duplication of the previous paragraph.

436: Note that there is some inflation of the Type I error rate due to multiple comparisons across response variables, as well. It is hard to draw a boundary on this and the authors have chosen an practical/acceptable rule to only adjust comparisons within response. However, it is easy to imagine an approach in the spirit of a 2-way ANOVA treating season and UV as factors.

Figure 6: This figure is helpful for seeing the lack of a clear-cut effect of UV. I would appreciate histograms of the data. Was the analysis robust to what looks like drastic violation of normality (particularly during spring)?

469: Repeating info from Methods.

493: It is really going out on a limb to say you see an increase in autumn bat detections with a P-value of 0.24. Would be fairer to re-arrange the statement into The response was positive but not significant.

637 and 697: Italics

542: For discussion: Would it be possible (within your experimental design) to introduce weather into a future analysis? I'm curious whether limiting analysis to days with permissive wind speeds, for instance, would unveil a stronger effect of UV.

Author Response

Author reconciliation of comments from Reviewer No. 2 of ‘Influencing activity of bats by dimly lighting wind turbine surfaces with ultraviolet light” by Cryan et al.  Author responses in blue.

( ) Extensive editing of English language and style required
( ) Moderate English changes required
(x) English language and style are fine/minor spell check required
( ) I don't feel qualified to judge about the English language and style

Yes

Can be improved

Must be improved

Not applicable

Does the introduction provide sufficient background and include all relevant references?

(x)

( )

( )

( )

Is the research design appropriate?

(x)

( )

( )

( )

Are the methods adequately described?

(x)

( )

( )

( )

Are the results clearly presented?

(x)

( )

( )

( )

Are the conclusions supported by the results?

( )

(x)

( )

( )

Comments and Suggestions for Authors

This study assesses the feasibility and efficacy of using dim UV light to deter bats from interacting with wind turbines, providing an important progress report on innovations in fatality reduction measures. The background and methodological description are thorough. An impressive amount of work went into the system design and data analysis. This being a preliminary study, the sample size is very limited. The authors used an appropriate statistical approach in their analysis and are careful to delineate the scope of inference they are able to draw.

Places for improvement in the manuscript include some repeated text and the opportunity for brevity.

We substantially revised the manuscript to be more concise, we removed repeated text, and reduced thematic elements that were tangential to the main flow of the narrative (e.g., relative merits of light over sound).

I would encourage the authors to check their presentation of the results to make sure they are being consistent in their treatment of non-significant results. Some are presented as suggestive of an effect while others are presented as the absence of effect. The criteria for this distinction should be made a bit more clear.

We appreciate this comment and have revised the text accordingly. We deleted more speculative phrases about non-significant results and we now include more direct statements about which results were statistically significant. We no longer refer to (or imply) increases or decreases in detection duration when statistics revealed no significance. The revision also includes clearer explanation of why the lack of statistical significance of risky behaviors with UV treatment is relevant in the context of statistically significant increases in observed activity.

It would also be helpful if the methods included some description of the signal provided by birds, bats, and insects during video analysis. What is the rough range of detection of each? What size object (perhaps in pixels?) was detectable by the frame-differencing algorithm used to flag periods of activity? The discussion notes that most insect observations were quite close to the cameras.

As detailed in the answer to a related comment by Reviewer 1, the revision now includes additional details of this process, including:

The automatic processing algorithm detected flying animals when their body heat triggered more than one pixel, which depended on animal size and distance from the camera. In our experience, visual classification of animals requires they are imaged at resolutions higher than approximately 8 pixels/m. We estimated maximum identification distances for this study of 20 m for medium-size insects like moths and beetles (~3.5 cm wingspan), 40 m for larger insects (~7.0 cm wingspan), 125 m for bats and similarly sized passerine birds (~20 cm wingspan), and about 150 m for the largest bats and large birds (e.g., ≥25 cm wingspan). Insects were visually discernable from bats and birds by differences in the clarity of image edges, characteristic shapes, and higher speeds at which they transited the video scenes. Bats and birds were visually discriminated by shape, wingbeat consistency, and flocking; animals observed soaring, flying with others in symmetrical formation, or showing a flap-glide flight cycle were assumed to be birds. We classified distant animals determined not to be insects (e.g., sharp image edges and transiting the video scene slowly over numerous video frames) but that were imaged at insufficient pixel densities for reliable identification as “high flyers.”

And,

We were not sure whether UV illumination increased bat activity through attraction to UV illumination independently of increased insect presence, but we might gather additional measures in future experiments to answer such questions. Because many of the medium-size insects (<3.5 cm wingspan) were detectable only up to about the height of where we mounted the lights on the turbine monopole (20 m) and larger insects (<7.0 cm wingspan) only about halfway up the monopole (40 m), the general detection area of in-sects was disproportionately lower to the ground than the larger detection area of bats. We did not try to estimate the heights of bats, or sample for insects near the highest reaches of the turbines where bats were discernable (125-150 m). We therefore cannot assess whether activity of insects and bats increased consistently near all parts of the turbine that were illuminated, or just at the lower reaches of the monopole near the lights.

Specific comments:

82: The use of triggers for operational changes is repeated from previous sentence.

This has been removed.

102: The justification for using light seems overwrought. Bats can see it from far away. Atmosphere and the speed of light? Probably minor points.

Agreed.  We removed many of the statements and redundancies about the benefits of light over sound in the revised version.  The focus is now mostly on light.

109: Can you shorten/combine this and the following paragraph? Also, the topic sentence here really belongs with the previous paragraph.

This was a very helpful suggestion. We considerably merged and condensed the paragraphs in question so now the focus is clearer, and the logic flows better.

137: It is unnecessary to label the study as speculative. The hypothesis that UV light could be a deterrent is well-supported and the need for new deterrent methods is convincing.

The revision retains reference to the assumption that bats visually mistake turbines in silhouette as trees as speculation, since we still don’t really know if that is the case. It would be hard to prove.

157: Structure this sentence to highlight effect on bats.

We appreciate this suggestion and restructured the sentence to read, “A field experiment to reduce Hawaiian hoary bat activity by flickering dim UV light on trees in a natural foraging environment demonstrated a decrease in bat vocalization and proximity to the illuminated area concomitant with a slight increase in insect activity [63].”

218: This and the next sentence include some repetition of information.

Revised to remove the repetition.

340: This is a duplication of the previous paragraph.

Fixed.

436: Note that there is some inflation of the Type I error rate due to multiple comparisons across response variables, as well. It is hard to draw a boundary on this and the authors have chosen an practical/acceptable rule to only adjust comparisons within response. However, it is easy to imagine an approach in the spirit of a 2-way ANOVA treating season and UV as factors.

Making multiple comparisons on the same data (e.g., separately testing for an increase and a decrease in bat counts in the spring period) runs the risk of inflating Type 1 error rates. In this study, we chose to examine the two “experiments” using a family-wise error rate. Given that we performed a suite of analyses on different datasets — taxa (e.g., birds, bats, insects), time periods (spring, autumn), and metrics (counts, duration) — there was no need to control error rate beyond that of the pair of tests examined (increasing or decreasing effect).

More stringent error control would have been appropriate had our hypothesis focused on whether there was a treatment effect on any of the groups examined. In that case, this could be achieved by collectively addressing the per-family error rate to limit the cumulative risk of false discovery. Although our study was exploratory and involved multiple post hoc tests, we did not make conclusions from an omnibus test of UV illumination and we limited inference to each independent dataset.

We do not believe that an ANOVA would be appropriate for our data. Auto-correlation very often arises in single-case designs, and classic ANOVA and approaches like least squares regression are not robust to violations of normality and homoscedasticity (Michiels and Onghena 2019). The nonparametric randomization test we performed within the single-case experimental design (SCED) framework avoided this pitfall.

Michiels, B.; Onghena, P. Randomized single-case AB phase designs: Prospects and pitfalls Behavior Research Methods 2019, 51, 2454-2476, doi:10.3758/s13428-018-1084-x.

Figure 6: This figure is helpful for seeing the lack of a clear-cut effect of UV. I would appreciate histograms of the data. Was the analysis robust to what looks like drastic violation of normality (particularly during spring)?

By necessity, our study was a single-case experiment (only data from one turbine was available) from which control-treatment responses were repeatedly observed. As such, analysis of the autocorrelated observations with standard parametric techniques was not appropriate. The ALIV (actual and linearly interpolated values) method and randomization test that we applied is a nonparametric technique that makes no distributional assumptions (Solmi et al. 2014), so histograms illustrating data distributions would not be relevant. We added text briefly explaining this to the methods section of the revision.

Solmi, F., Onghena, P., Salmaso, L. and Bulté, I., 2014. A permutation solution to test for treatment effects in alternation design single-case experiments. Communications in Statistics-Simulation and Computation43(5), pp.1094-1111.

469: Repeating info from Methods.

Removed redundancies and streamlined between methods and discussion.

493: It is really going out on a limb to say you see an increase in autumn bat detections with a P-value of 0.24. Would be fairer to re-arrange the statement into The response was positive but not significant.

We agree and have made the suggested edit. The sentence now reads:

Likewise, although count and duration of bat detections during the autumn period trend-ed positive with UV treatment, the observed differences were not significant (counts: ALIV mean difference = 6.7; BH p-value = 0.214; duration: ALIV mean difference = 66.7; BH p-value = 0.243).”

637 and 697: Italics

Scientific names all italicized in revision.

542: For discussion: Would it be possible (within your experimental design) to introduce weather into a future analysis? I'm curious whether limiting analysis to days with permissive wind speeds, for instance, would unveil a stronger effect of UV.

This is a good question and one we considered both during study design and analysis, but the dilemma of a small sample size prevented us from being able to investigate weather effects. As a nonparametric method, the single-case experimental design (SCED) is not suitable for analyses aimed at incorporating multiple predictor variables. It is focused entirely on testing the effect of a single variable; that is, the application of a particular treatment and its comparison to a control. At the cost of limited inference as to the causal effect of additional factors, the method’s simplicity avoids the constraints of parametric tests (i.e., distributional assumptions) and provides an approach for straightforward assessment of treatment effect for single-case subjects. Limited but effective!

(SEE ATTACHED PDF FOR COPIES OF THESE RESPONSES TO REVIEWERS AND A MARKED COPY OF THE MS SHOWING CHANGES WE MADE IN THE REVISION)

Round 2

Reviewer 1 Report

Given the low sample size of n=1 this is a paper for a short communication and not for a full article. The inference we can draw from this paper are very limited based on a single turbine studied, lacking any replicates. The intrinsic flaw of detecting flying animals in various focal areas that do not necessarily overlap, e.g. bats and insects, adds another layer of uncertainty to this study. This study may help other researchers to design a replicate study thoroughly, but it does not help to answer the question of whether UV stroboscopic light may deter bats or not. Therefore, the discussion should focus on how to improve the study design and not on biological questions which remain unanswered. Some of my previous comments were not adequately addressed from my point of view which suggests to me that the authors are unwilling to see the limitations of their study.

Author Response

Given the low sample size of n=1 this is a paper for a short communication and not for a full article. The inference we can draw from this paper are very limited based on a single turbine studied, lacking any replicates.

We maintain our belief that the single-case statistical framework we used is valid and that we can infer meaningful patterns from the study.  If the reviewer is uncomfortable with this valid method, we agree to disagree. We report a statistically significant probability that insects and bats were attracted to the wind turbine when it was experimentally UV illuminated—that alone could interest many readers.  Ambiguity arises when we try to interpret what we consider as the fact of higher insect and bat activity with UV at this single turbine during the time we studied it.  We do not believe readers would benefit from a short communication focused on improved study design (suggested in subsequent comment).  A briefer treatment could not include our main finding of UV attracting insects and bats coupled with the necessary caveats and discussion on caution about inferring too much from a limited sample and the unintended effects and animal welfare concerns.  We also believe we present important perspective relevant to other visual-based deterrence experiments and that these methods, and lessons learned, can be effectively applied to tests of other types of deterrence technologies.  

The intrinsic flaw of detecting flying animals in various focal areas that do not necessarily overlap, e.g. bats and insects, adds another layer of uncertainty to this study.

Contrary to this opinion, we strongly disagree with the assessment that unequal detectability of different kinds of animals with thermal cameras is an intrinsic flaw.  We developed, tested, and used this method because we believe it is the best available observation technique for studying and quantifying the activity and detailed behaviors of various taxa of flying animals close to wind turbines.  The alternatives, including acoustic monitoring and carcass ground searches, simply do not compare in terms of detectability and feasibility.  If the reviewer has suggestions for alternative methods of detecting multiple, diverse taxa of flying animals at different distances that are independent of animal size or environmental conditions, we will consider them carefully.  The various focal areas and detection distances of the thermal cameras do indeed overlap, with the exception that insects are not easily detected above about 20-40m.  Birds and bats are detectable and discernable from insects and each other from the highest reaches of the wind turbine all the way down to the ground.  Despite this variable detection range for different taxa, we do not believe that we ever overstep the bounds of inference when analyzing and interpreting the data: we do not directly compare bat, bird, high flyer, and insect activity measures among taxa.  The statistical method we used only made comparisons within each group (e.g., bats, birds, insects). We specifically removed much text from the manuscript detailing the limitations of acoustic monitoring/deterrence in response to prior review comments, so considering the lack of alternative methods for this type of data collection, we are at a loss for a less flawed method. 

This study may help other researchers to design a replicate study thoroughly, but it does not help to answer the question of whether UV stroboscopic light may deter bats or not.

Yes, we agree with this comment and believe the manuscript as written effectively addresses the fact that we remain uncertain about whether UV light deters bats.  Science is an incremental process, and this was the first time that this hypothesis had been tested. Our results offer novel insight into how to design a relevant study for this type of applied deterrence research. For example, the statistical tests that circumstances forced us to eventually use were specifically chosen because they accounted for positive and negative responses to treatment, whereas without considering the possibility of attraction future researchers assessing deterrence devices might choose a study design that only accounts for decreases (e.g., one-tailed statistical tests). We believe it is our responsibility as scientists to report results in a disinterested way, regardless of outcome, such that the reader clearly understands in this particular case that we most definitely do not know the answer to whether UV deters bats.

Therefore, the discussion should focus on how to improve the study design and not on biological questions which remain unanswered.

Please see our response to first comment.

Some of my previous comments were not adequately addressed from my point of view which suggests to me that the authors are unwilling to see the limitations of their study.

We fully admit to the limitations of our study here and in the manuscript.  We are very willing to further elaborate on the limitations of our study should the editors see that as an appropriate direction and provide guidance as to where and how. If the reviewer has additional points that we did not adequately address in our previous comments, or in this subsequent round, please let us know the specifics and we will do our best to reconcile them.